# INFORMATION PLANE ANALYSIS FOR DROPOUT NEURAL NETWORKS

**Linara Adilova**
Ruhr University Bochum,
Faculty of Computer Science
linara.adilova@ruhr-uni-bochum.de

**Bernhard C. Geiger**
Know-Center GmbH
geiger@ieee.org

**Asja Fischer**
Ruhr University Bochum,
Faculty of Computer Science
asja.fischer@ruhr-uni-bochum.de

## ABSTRACT

The information-theoretic framework promises to explain the predictive power of neural networks. In particular, the information plane analysis, which measures mutual information (MI) between input and representation as well as representation and output, should give rich insights into the training process. This approach, however, was shown to strongly depend on the choice of estimator of the MI. The problem is amplified for deterministic networks if the MI between input and representation is infinite. Thus, the estimated values are defined by the different approaches for estimation, but do not adequately represent the training process from an information-theoretic perspective. In this work, we show that dropout with continuously distributed noise ensures that MI is finite. We demonstrate in a range of experiments[1] that this enables a meaningful information plane analysis for a class of dropout neural networks that is widely used in practice.

## 1 INTRODUCTION

The information bottleneck hypothesis for deep learning conjectures two phases of training feed-forward neural networks (Shwartz-Ziv and Tishby, 2017): the fitting phase and the compression phase. The former corresponds to extracting information from the input into the learned representations, and is characterized by an increase of mutual information (MI) between inputs and hidden representations. The latter corresponds to forgetting information that is not needed to predict the target, which is reflected in a decrease of the MI between learned representations and inputs, while MI between representations and targets stays the same or grows. The phases can be observed via an information plane (IP) analysis, i.e., by analyzing the development of MI between inputs and representations and between representations and targets during training (see Fig. 1 for an example). For an overview of information plane analysis we refer the reader to (Geiger, 2022).

While being elegant and plausible, the information bottleneck hypothesis is challenging to investigate empirically. As shown by Amjad and Geiger (2020, Th. 1), the MI between inputs and the representations learned by a deterministic neural network is infinite if the input distribution is continuous. The standard approach is therefore to assume the input distribution to be discrete (e.g., equivalent to the empirical distribution of the dataset $S$ at hand) and to discretize the real-valued hidden representations by binning to allow for non-trivial measurements, i.e., to avoid that the MI always takes the maximum value of $\log(|S|)$ (Shwartz-Ziv and Tishby, 2017). In this discrete and deterministic setting the MI theoretically gets equivalent to the Shannon entropy of the representation. Considering the effect of binning, however, the decrease of MI is essentially equivalent to geometrical compression (Basirat et al., 2021). Moreover, the binning-based estimate highly depends on the chosen bin size (Ross, 2014). To instead work with continuous input distributions, Goldfeld

---

[1]Code for the experiments is public on https://github.com/link-er/IP_dropout.

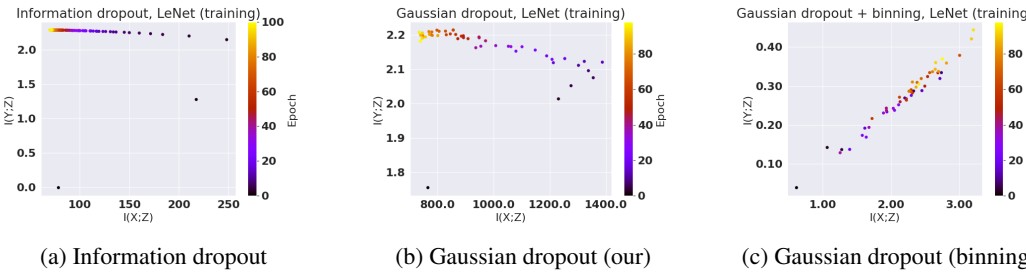

Figure 1: IPs w.r.t. the activations of one layer with information dropout or Gaussian dropout in a LeNet network. In contrast to the IP based on estimating MI using binning, our estimates (both for Gaussian and information dropout) clearly show compression. This suggests that even if MI is finite, the binning estimator fails to converge to the true MI (see also Section 4).

et al. (2019) suggest to replace deterministic neural networks by stochastic ones via adding Gaussian noise to each of the hidden representations. This kind of stochastic networks is rarely used in practice, which limits the insights brought by the analysis.

In contrast, dropout, being a source of stochasticity, is heavily used in practice due to its effective regularizing properties. The core questions investigated in this work therefore are: i) Can we obtain accurate and meaningful MI estimates in neural networks with dropout noise? ii) And if so, do IPs built for dropout networks confirm the information bottleneck hypothesis? Our main contributions answer these questions and can be summarized as follows: We present a theoretical analysis showing that binary dropout does not prevent the MI from being infinite due to the discrete nature of the noise. In contrast, we prove that dropout noise with any continuous distribution not only results in finite MI, but also provides an elegant way to estimate it. This in particular holds for Gaussian dropout, which is known to benefit generalization even more than binary dropout (Srivastava et al., 2014), and for information dropout (Achille and Soatto, 2018). We empirically analyze the quality of the MI estimation in the setup with Gaussian and information dropout in a range of experiments on benchmark neural networks and datasets. While our results do not conclusively confirm or refute the information bottleneck hypothesis, they show that the IPs obtained using our estimator exhibit qualitatively different behavior than the IPs obtained using binning estimators and strongly indicate that a compression phase is indeed happening.

## 2 MUTUAL INFORMATION ESTIMATION FOR NEURAL NETWORKS

We use the following notation: Lower-case letters denote realizations of random variables (RVs), e.g., $b$ denotes a realization of the RV $B$; $H(A)$ denotes the Shannon entropy of a discrete RV $A$ whose distribution is denoted $p_a$; $h(B)$ is the differential entropy of a continuous RV $B$ whose distribution is described by the probability density function $p_b$; $I(A; B)$ is the MI between RVs $A$ and $B$; $X \in \mathcal{X} \subseteq \mathbb{R}^n$ and $Y \in \mathcal{Y}$ are the RVs describing inputs to a neural network and corresponding targets; $f(X)$ is the result of the forward pass of the input through the network to the hidden layer of interest; $Z$ is an $N$-dimensional RV describing the hidden representations.

The caveats of different approaches to measure the MI between input $X$ and hidden representation $Z$ of a neural network – e.g., the MI being infinite for deterministic neural networks and continuous input distributions, the dependence of the MI estimate on the parameterization of the estimator, etc. – were discussed widely in the literature (Saxe et al., 2019; Geiger, 2022) and are briefly reviewed in this section. These caveats do not appear for the MI measured between representations $Z$ and targets $Y$, since the target is in most cases a discrete RV (class), for which MI is always finite.

One option for estimating $I(X; Z)$ is to assume the input to be drawn from a discrete distribution. This view is supported by the finiteness of the accuracy of the used computational resources (Lorenzen et al., 2021) and makes it easy to use a finite dataset $S$ to describe the distribution. In such setup, the distribution of $(X, Y)$ is assumed uniform on the dataset $S$, and the discretization of $Z$ is performed at a fixed bin size (e.g., corresponding to the computer precision). The MI between

$X$ and the discretized $\hat{Z}$ is computed as $I(X; \hat{Z}) = H(\hat{Z}) - H(\hat{Z}|X) = H(\hat{Z}) - 0 = H(\hat{Z})$, where $H(\hat{Z}|X) = 0$ since $f(\cdot)$ and the discretization of $Z$ are deterministic. Thus, the estimated MI between input and representation corresponds to the entropy of the discretized representation, which for small bin sizes is equal to the entropy $H(X) = \log|S|$ of the empirical distribution on the dataset, unless $f(\cdot)$ maps different points from the dataset to the same point in latent space.

A different option that is more aligned to the common description of real-world data is to assume $X$ to be drawn from a continuous distribution. If the network transformation $f(\cdot)$ results in a discrete distribution of the representations $Z$, one can use the decomposition $I(X, Z) = H(Z) - H(Z|X) = H(Z)$ to estimate MI based on Shannon entropy, provided that the sample size is sufficiently large (note that the dimensionality $N$ of $Z$ may be large, and therefore the estimation of $H(Z)$ may suffer from the curse of dimensionality). However, as shown in Theorem 1 of (Amjad and Geiger, 2020) for neural networks with commonly used activation functions the distribution of the latent representation is not discrete. In this case (i.e., $f(\cdot)$ is deterministic, $X$ is continuous, and $Z$ is not purely discrete) the MI between $X$ and $Z$ is infinite[2]. By binning, i.e., by quantizing $Z$ to a discrete RV $\hat{Z}$, the MI $I(X; \hat{Z}) = H(\hat{Z})$ remains finite, but the qualitative behavior of this entropy will be defined by properties of activation functions and selected bin size (Saxe et al., 2019).

From the discussion above it follows that estimating $I(X; Z)$ in deterministic neural networks is an ill-posed problem, and that the estimates reveal not an information-theoretic picture, but often rather a geometric one that is determined by the properties of the chosen estimators. As a solution to the aforementioned challenges, several authors have suggested to investigate the information planes of stochastic neural networks instead (Amjad and Geiger, 2020; Goldfeld et al., 2019). Goldfeld et al. (2019) proposed to add zero-mean Gaussian noise $D$ to the representations during training. This transforms a deterministic neural network into a stochastic one that was shown to yield similar training results and predictive abilities of the model. The addition of Gaussian noise in $Z = f(X) + D$ guarantees a finite MI[3] and therefore allows for estimating MI using Monte Carlo sampling with bounds on the estimation error. Futhermore, it links the information-theoretic perspective of the IP to geometric effects taking place in latent space. Indeed, when the MI between input and representation is decreasing, it means that noise-induced Gaussians centered at the representations of different data points overlap more strongly. Thus, it is becoming harder to distinguish between inputs of the same class based on their representations, which translates into lower MI between representation and input while leaving MI between representation and target unchanged.

As discussed above, for continuous input distributions both the IPs of deterministic neural networks as well as of stochastic neural networks with additive noise show a geometric picture (and in the former case the geometric interpretation is the only valid one, since MI is infinite in this case). Therefore, in this work we study the estimation of MI in networks with dropout layers, i.e., in settings where the stochasticity is introduced by multiplicative, rather than additive noise. In what follows we will investigate the requirements on the multiplicative noise for MI to remain finite, and whether the resulting IPs confirm the information bottleneck hypothesis.

## 3 Mutual Information in Dropout Networks

As discussed in the previous section, the MI between inputs and hidden representations of deterministic networks is infinite, if we assume the input distribution to be continuous. To overcome this problem, some form of stochasticity has to be introduced. While adding noise to activations (Goldfeld et al., 2019) indeed allows to compute the MI, this is not used in most contemporary neural networks. In contrast, neural networks with dropout are one of the most popular classes of neural networks used in practice and are stochastic in nature as well: Adding a dropout layer to a neural network corresponds to multiplying the hidden representation with some form of random noise. Formally, denoting the random noise by a RV $D$ of the same dimension as $f(X)$, the hidden representation becomes $Z = f(X) \circ D$, where $\circ$ denotes element-wise multiplication. In the most basic form, $D$ follows a Bernoulli distribution (Srivastava et al., 2014). Such binary dropout is widely used and can intuitively been understood as "turning off" a fraction of neurons during training. There is a

---

[2]There are multiple mathematical derivations explaining why MI is infinite, one for example is discussed in (Saxe et al., 2019, Appendix C).

[3]At least when the $p_x$ and $f(\cdot)$ are such that $f(X)$ has finite variance, then the finiteness of MI follows from the result about the capacity of the additive Gaussian noise channel, cf. (Cover and Thomas, 1991, eq. (10.17)).

variety of other dropout schemes, including multiplicative Gaussian noise, fast dropout (Wang and Manning, 2013), or variational dropout (Kingma et al., 2015). Information dropout (Achille and Soatto, 2018) is a variant that uses a closed-form expression of MI as regularization term. In order to obtain such closed form, dropout noise is sampled from a log-normal distribution, and the prior distribution on representations is chosen depending on the activation function (ReLU or Softplus). We provide details on the derivation in Appendix A.1.

In this section, we investigate whether neural networks with dropout have indeed finite MI between input $X$ and representation $Z$. While we first show a negative result by proving that binary dropout still leads to $I(X; Z) = \infty$, our Theorem 3.3 shows that dropout with continuous distribution keeps MI finite. This fact allows us to estimate MI for such dropout neural networks in Sections 4 and 5.

## 3.1 Binary Dropout

We start by analyzing binary dropout, which forces individual neurons to be "turned off" with some probability. More formally, the output of each neuron is multiplied with an independent Bernoulli RV that is equal to 1 with a predefined probability $p$. The following theorem shows that this kind of (combinatorial) stochasticity is insufficient to prevent $I(X; Z)$ from becoming infinite.

**Theorem 3.1.** *In the setting of (Amjad and Geiger, 2020, Th. 1), let the output $f(\cdot)$ of a hidden layer be parameterized as a deterministic neural network with $\hat{N}$ neurons, let $B \in \{0, 1\}^{\hat{N}}$ be the set of independent Bernoulli RVs characterizing the dropout pattern, and let $Z = f_B(X)$ denote the output of the hidden layer after applying the random pattern $B$. Then it holds that $I(X; Z) = \infty$.*

In the proof (provided in Appendix A.2) we use the fact that dropout mask $b = (1, 1, \ldots, 1)$ leads to an infinite MI. While the Bernoulli distribution guarantees that $b = (1, 1, \ldots, 1)$ always has non-zero probability, other distributions over $\{0, 1\}^{\hat{N}}$ might not have this property. Theorem 3.1 can however be generalized to arbitrary distributions over $\{0, 1\}^{\hat{N}}$:

**Theorem 3.2.** *In the setting of (Amjad and Geiger, 2020, Th. 1), let the output $f(\cdot)$ of a hidden layer be parameterized as a deterministic neural network with $\hat{N}$ neurons, let $B \in \{0, 1\}^{\hat{N}}$ be the binary random vector characterizing the dropout pattern, and let $Z = f_B(X)$ denote the output of the hidden layer after applying the random pattern $B$. Then, it either holds that $I(X; Z) = \infty$ or that $I(X; Z) = 0$ if the dropout patterns almost surely disrupt information flow through the network.*

The proof for the theorem is provided in Appendix A.3.

Both Theorem 3.1 and Theorem 3.2 cover as a special case the setting where dropout is applied to only a subset of layers, by simply setting those elements of $B$ to 1 that correspond to a neuron output without dropout. If dropout is applied to only a single layer, then $f_B(X) = f(X) \circ B'$, where $B'$ is the dropout pattern of the considered layer and $\circ$ denotes the element-wise product.

As a consequence of Theorem 3.2, for neural networks with binary dropout any finite estimate of MI is "infinitely wrong", and the resulting IP does not permit an information-theoretic interpretation. Essentially, the stochasticity added by binary dropout is combinatorial, and hence cannot compensate the "continuous" stochasticity available in the input $X$.

## 3.2 Dropout with Continuous Noise

As proposed by Srivastava et al. (2014), dropout can also be implemented using continuous Gaussian noise with mean vector $\mu = \mathbf{1}$ and diagonal covariance matrix $I\sigma^2$ with fixed variance $\sigma^2$. Achille and Soatto (2018), in contrast, proposed log-normally distributed dropout noise, the variance of which depends on the input sample $x$ (this is termed information dropout). Generalizing both Gaussian and information dropout, in this section we consider continuously distributed multiplicative noise $D$. In contrast to binary noise sampled from a discrete distribution, continuously distributed noise turns the joint distribution of $(Z, X)$ to be absolutely continuous with respect to the marginals of $Z$ and $X$ allowing for finite values of MI between the input $X$ and the hidden representation $Z$. The following theorem states that the MI between input and the hidden representation of the dropout layer is indeed finite even if the variance of the noise depends on the input.

**Theorem 3.3.** *Let $X$ be bounded in all dimensions, $f(\cdot)$ be parameterized by a deterministic neural network with Lipschitz activation functions, and let $Z = f(X) \circ D(X)$, where the components of*

*noise $D(X) = (D_1(X), \ldots, D_N(X))$ are conditionally independent given $X$ and have essentially bounded differential entropy and second moments, i.e., $\mathbb{E}[D_i(X)^2] \leq M < \infty$ $X$-almost surely, for some $M$ and all $i = 1, \ldots, N$. Then, if the conditional expectation $\mathbb{E}[\log(|f(X)|) \mid |f(X)| > 0]$ is finite in each of its elements, we have $I(X; Z) < \infty$.*

Theorem 3.3 (proof in Appendix A.4) can be instantiated for Gaussian dropout, where $D_i(x) = D_i \sim \mathcal{N}(1, \sigma^2)$, and for information dropout, where $D_i(x) \sim \log \mathcal{N}(0, \alpha^2(x))$. Note that for information dropout we have to ensure that the (learned) variance $\alpha^2(x)$ stays bounded from above and below; e.g., in the experiments of Achille and Soatto (2018), $\alpha^2(x)$ is restricted to be below 0.7.

The requirement that the conditional expectation $\mathbb{E}[\log(|f(X)|) \mid |f(X)| > 0]$ is finite in each of its elements is critical for the proof. Indeed, one can construct a synthetic (albeit unrealistic) example for which this condition is violated:

**Example 3.4.** *Let $X'$ have the following probability density function*

$$p_{x'}(x') = \begin{cases} 2^{-n}, & \text{if } x' \in [2^n, 2^n + 1), n = 1, 2, \ldots \\ 0, & \text{else} \end{cases}$$

*Evidently, $\mathbb{E}[X'] = \infty$. Then, $X = e^{-X'}$ is bounded, since its alphabet is a subset of $(0, e^{-2}]$.*

*Now consider a neural network with a single hidden layer with one neuron. Let the weight from $X$ to the single neuron be $1$, and assume that the neuron uses a ReLU activation function. Then,*

$$\mathbb{E}[\log |f(X)|] = \mathbb{E}[\log |X|] = \mathbb{E}[\log |e^{-X'}|] = \mathbb{E}[-X'] = -\infty \ .$$

It can be shown that in this example the probability density function of $X$ (as well as of $f(X)$) is not bounded. Under the assumption that the probability density function $p_f$ of $f(X)$ is bounded, the conditional expectation in the assertion of the theorem is finite: Assuming that $p_f \leq C < \infty$, by the law of unconscious statistician we have

$$
\begin{aligned}
\mathbb{E}_x[\log(|f(X)_i|) \mid |f(X)_i| > 0] &= \int_0^{\|f(X)_i\|_\infty} \log(f) p_f(f) \mathrm{d}f \\
&= \underbrace{\int_0^1 \log(f) p_f(f) \mathrm{d}f}_{I_1} + \underbrace{\int_1^{\|f(X)_i\|_\infty} \log(f) p_f(f) \mathrm{d}f}_{I_2} \ .
\end{aligned}
$$

It is obvious that $I_2$ is positive and finite. Due to the boundedness of $p_f$ we also have $I_1 \geq C \int_0^1 \log(f) \mathrm{d}f = C f(\log(f) - 1)|_0^1 = -C > -\infty$.

However, the boundedness of $p_f$ of is hard to guarantee for an arbitrary neural network. In contrast, the boundedness of $p_x$ is more realistic and easier to check. For bounded $p_x$ we can prove (in Appendix A.5) the finiteness of the expectation $\mathbb{E}[\log(|f(X)|) \mid |f(X)| > 0]$ for ReLU networks:

**Proposition 3.5.** *Consider a deterministic neural network function $f(\cdot)$ constructed with finitely many layers, a finite number of neurons per layer, and ReLU activation functions. Let $X$ be a continuously distributed RV with probability density function $p_x$ that is bounded ($p_x \leq P < \infty$) and has bounded support $\mathcal{X}$. Then, the conditional expectation $\mathbb{E}[\log(|f(X)|) \mid |f(X)| > 0]$ is finite in each of its elements.*

Finally, note that Theorem 3.3 assumes that the network is deterministic up to the considered dropout layer. This does not come with a loss of generality for feed-forward networks (e.g., with no residual connections): Indeed, one can apply Theorem 3.3 to the first hidden layer representation $Z^{(1)}$ with dropout, where this assumption always holds. Then, for the $\ell$-th hidden layer and irrespective of whether this layer also has dropout, the MI $I(X; Z^{(\ell)})$ is finite due to the data processing inequality (Cover and Thomas, 1991, Th. 2.8.1). Therefore, Theorem 3.3 ensures that MI is finite for all hidden layers after the first continuous dropout layer.

## 4 ESTIMATION OF MI UNDER CONTINUOUS DROPOUT

We now consider estimating $I(X; Z)$ in networks with continuously distributed dropout, starting with information dropout. As discussed by Achille and Soatto (2018), networks with information

dropout are trained with the cross-entropy loss $\ell_{ce}$ (which is involved in the known variational lower bound $I(Z;Y) \geq H(Y) - \ell_{ce}$) and regularized using a variational upper bound on $I(X;Z)$. Therefore, estimates of the quantities displayed in the information plane are directly used in the training loss and, thus, easy to track, at least for softplus activation functions[4].

In the case of Gaussian dropout, to estimate $I(X;Z)$ we approximate $h(Z)$ and $h(Z|X)$ separately (pseudocode is given in Algorithm 1 in Appendix A.6).

For estimating $h(Z)$ we employ a Monte Carlo (MC) estimate, similar to the one proposed by Goldfeld et al. (2019). That is, we approximate the distribution of $Z$ as a Gaussian mixture, where we draw samples $f(x^{(j)}), j = 1, \ldots, |S|$ and place Gaussians with a diagonal covariance matrix with variances $\sigma^2 |f(x^{(j)})_i|^2, i = 1, \ldots, N$ on each sample $f(x^{(j)})$. For a sanity check, we also compute an upper bound of $h(Z)$ given by the entropy of a Gaussian with the same covariance matrix as $Z$. Note that the estimation of the upper bound requires a sufficiently large number of samples to guarantee that the sample covariance matrix is not singular and that the resulting entropy estimate is finite.

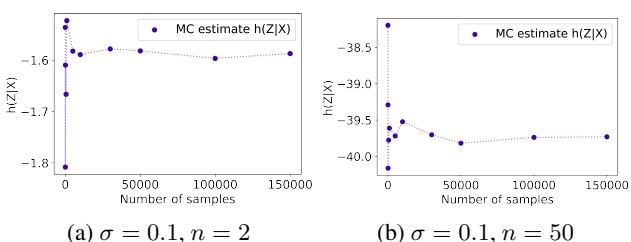

(a) $\sigma = 0.1, n = 2$      (b) $\sigma = 0.1, n = 50$

Figure 2: Independent of the dimensionality, MC estimation of $h(Z|X)$ stabilizes with increasing amount of samples.

For each fixed $x$ the conditional distribution $p_{z|x}$ is a Gaussian distribution $\mathcal{N}(f(x), \text{diag}(\{\sigma^2 |f(x)_i|)^2\}))$. Moreover, when the input is fixed, the components of $Z|X = x$ are independent, since components of the noise are independent. This allows to compute $h(Z|X)$ as a sum of $h(Z_i|X)$ where $Z_i$ is the $i$-th component of the representation vector. The computation of $h(Z_i|X)$ requires integration over the input space for computing the mathematical expectation $\mathbb{E}_x[h(Z_i|X = x)]$. This can be approximated via MC sampling. That is, we approximate $h(Z_i|X)$ by $1/|S| \sum_{j=1}^{|S|} h(Z_i|X = x^{(j)})$ where $h(Z_i|X = x^{(j)}) = \log(|f(x^{(j)})_i|\sigma\sqrt{2\pi e})$.

(a) $\sigma = 0.1, n = 1$      (b) $\sigma = 0.1, n = 50$

Figure 3: Estimates of the differential entropy $h(Z)$ of the hidden representation $Z$. With growing dimensionality of $X$, the Gaussian upper bound becomes very loose, compared to the Gaussian mixture-based MC estimation.

We consider a simple toy problem for validating our approach to estimating MI: the input $X$ is generated from an $n$-dimensional standard normal distribution, modified with a function $f(X) = 2X + 0.5$, and then subjected to Gaussian dropout distributed according to $\mathcal{N}(1, \sigma^2)$. We investigate the convergence of our estimator for $h(Z|X)$ for increasing number of samples. For each input data point, we generate 10 noise masks, thus obtaining 10 samples of $Z$ for each $x^{(j)}$. The results in Fig. 2 show that the estimation stabilizes with larger amount of samples for different dimensionality of the data. We also compare the estimate to the upper bound for $h(Z)$ in Fig 3.

We finally compare our estimation of MI to binning, the EDGE estimator (Noshad et al., 2019), and the lower bounds analyzed by McAllester and Stratos (2020). The results are shown in Fig. 4. In the plot, doe stands for the difference-of-entropies (DoE) estimator and doe_l stands for DoE with logistic parametrization (McAllester and Stratos, 2020). The binning estimator underestimates the

---

[4]Indeed, for softplus activation functions, the variational approximation of $I(X;Z)$ is available in closed form, while for ReLU activation functions, the available expression is only useful for minimizing, rather than for computing, $I(X;Z)$ (see Appendix A.1).

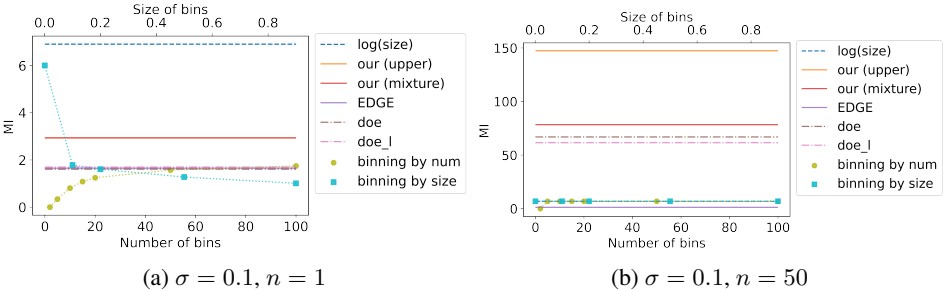

(a) $\sigma = 0.1$, $n = 1$  (b) $\sigma = 0.1$, $n = 50$

Figure 4: Comparison of various approaches to MI estimation for the toy example with multiplicative Gaussian noise. For low-dimensional $X$ and $Z$, different bin sizes lead to different MI estimates of the binning estimator. For higher dimensions, the binning-based estimate is collapsing. Our estimation is very close to the lower bound estimation proposed by McAllester and Stratos (2020), while still being larger as expected.

MI when the bin size is large and overestimates it with small bin size (Ross, 2014), which can be clearly seen in the plots where bins are organized both by size (upper axis) and by number (lower axis). Moreover, with the high-dimensional data, binning hits the maximal possible value of $\log(|S|)$ very fast, not being able to reach larger MI values. According to McAllester and Stratos (2020), lower bound-based MI estimators (e.g., MINE (Belghazi et al., 2018)) also need exponentially (in the true value of MI) many data points for a good value approximation, otherwise they will always heavily underestimate the MI.

Further plots for different dropout variances and inputs dimensionality are given in Appendix A.6.

## 5 INFORMATION PLANE ANALYSIS OF DROPOUT NETWORKS

We use the estimators described in the previous section for an IP analysis of networks with Gaussian and information dropout. We always consider only the representation corresponding to the first dropout layer [5] and measure the MI in nats, e.g., use the natural logarithm. For estimating $I(Y; Z)$, we employ the EDGE estimator (Noshad et al., 2019) for Gaussian dropout and variational estimate for information dropout. IPs created using the binning estimator use binning for both $I(X; Z)$ and $I(Y; Z)$.

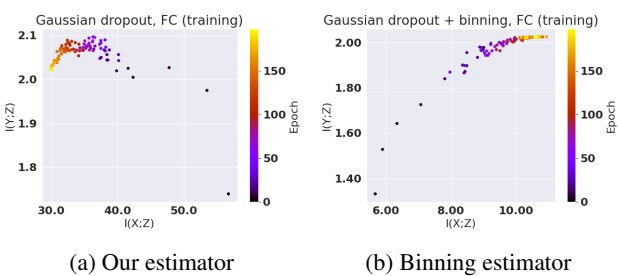

(a) Our estimator  (b) Binning estimator

Figure 5: IPs for a FC network with Gaussian dropout trained on MNIST. Compared to the binning estimation of MI our approach shows compression.

In the first set of experiments we investigate the difference between IPs obtained via our proposed estimator and via binning. The analysis on the MNIST dataset was performed for a LeNet network (LeCun et al., 1998) that achieves 99% accuracy and a simple fully-connected (FC) network with three hidden layers ($28 \times 28 - 512 - 128 - 32 - 10$) and softplus activation functions achieving 97% accuracy. We analyze both information dropout and Gaussian dropout in the LeNet network and only Gaussian dropout in the FC network. In both cases dropout is applied on penultimate layers. We compare IPs based on binning estimators to IPs based on our estimators in Fig. 1 and Fig. 5.

---

[5]This makes the MI estimation more efficient, since the previous part of the network is deterministic which allows for an analytical expression of $h(Z|X = x)$. Note however, that the estimation could be extended to higher layers as well since for those MI also remains finite. However, an estimator different from ours should be used for those layers.

We also analyze the IPs for a ResNet18 trained on CIFAR10 (see Fig. 6), where we added an additional bottleneck layer with 128 neurons and Gaussian dropout before the output layer, and which achieves an accuracy of 94%.

Interestingly, for all networks and datasets we observe significant compression for our estimator and a lack of compression for binning estimators (also for different bin size, see Appendix A.8). This indicates that either the MI compression measured in dropout networks is different from purely geometrical compression, or

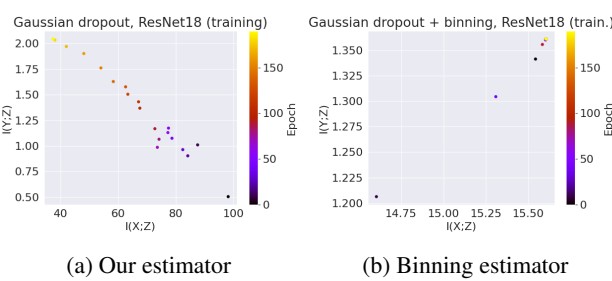

(a) Our estimator      (b) Binning estimator

Figure 6: IPs for a ResNet18 network with Gaussian dropout trained on CIFAR10. In contrast to the binning-based estimator of MI our approach clearly shows compression.

that the number of samples $|S|$ is insufficient to reliably estimate $I(X; Z)$ by binning.

In the second set of experiments, we analyze IPs in information dropout networks, with MI estimations as described before. To this end, we trained a fully convolutional neural network (fullCNN) on CIFAR10 using code provided by Achille and Soatto (2018). Training proceeded for 200 epochs using SGD with momentum and, different from the original setup, with only one dropout layer after the third convolutional layer. The batch size was set to 100, the learning rate was initially set to 0.05 and was reduced by multiplying it with 0.1 after the 40, 80, and 120 epoch. The network was trained with different values of the regularization weight $\beta$ and different amounts of filters in the convolutional layers. That is, the full-size fullCNN has 3 layers with 96 filters succeeded by 4 layers with 192 filters, while only 25% of these filters are constituting the small network. Also different from the original setup, we allowed the noise variance to grow up to 0.95 in order to see the effect of the limited

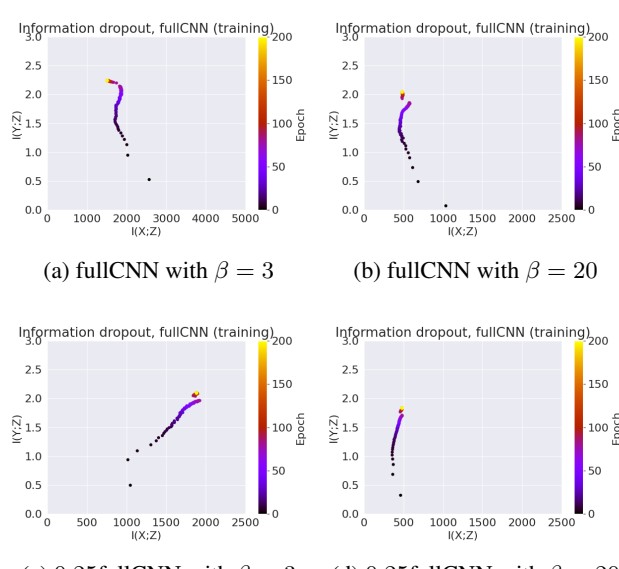

(a) fullCNN with $\beta = 3$      (b) fullCNN with $\beta = 20$

(c) 0.25fullCNN with $\beta = 3$    (d) 0.25fullCNN with $\beta = 20$

Figure 7: IPs demonstrate more (a), (c) and less (b), (d) compression of MI between input and representation depending on $\beta$. The values of $I(X; Z)$ are smaller for the smaller network (c) and (d).

information between representation and input more pronounced. Results are shown in Fig. 7. It can be seen that regularizing $I(X; Z)$ is effective (i.e., larger values of $\beta$ lead to smaller $I(X; Z)$), and that regularizing too strongly ($\beta = 20$) leads to worse performance: the test error is 5% higher and train error is 10% higher. We can further see stronger compression for smaller $\beta$ and almost no compression for larger $\beta$. We conjecture that compression can only become visible if sufficient information is permitted to flow through the network (which happens only for small $\beta$). Fig. 7 (c) and (d) show the IPs for the small fullCNN. It can be seen that the smaller network appears not to compress at all (see Fig. 7 (c)), but that $I(X; Z)$ rather increases throughout training until it is at the same level as in Fig. 7 (a). This indicates that $\beta$ determines to which point in the IP information compresses, and that the IP curve that is traversed during training depends on the overall capacity of the neural network.

Plots for the additional experiments can be found in Appendix A.8.

## 6 DISCUSSION

Whether or not information-theoretic compression is correlated with improved generalization is the main question connected to and the most prominent justification for information plane analysis of deep neural networks. Such a connection, however, can only be tested for neural networks for which MI is finite and therefore measurable. In our theoretical analysis, we investigate if different variants of dropout noise allow for finite values of MI under an assumption of a continuous input distribution. We answered this question positively by showing that in networks with certain constraints on the induced distribution of the representations, continuous dropout noise with finite differential entropy prevents $I(X; Z)$ from becoming infinite. We have further shown that these constraints on the distribution of the representation are satisfied in ReLU networks if the probability density function of the input is bounded.

Following this conclusion we propose an MC-based estimate of MI in Gaussian dropout networks and perform an IP analysis for different networks with Gaussian and information dropout on different datasets. The experiments show that the binning estimator behaves very differently from our estimator: While our estimator mostly exhibits compression in the IP, the binning estimator does not. Further, the values of $I(X; Z)$ for our estimator are often orders of magnitude larger than the values of $I(Y; Z)$, especially when compared to the binning estimator. Assuming that the proposed estimators are reasonably accurate, this makes a connection between information-theoretic compression and generalization questionable. While these preliminary experiments do not conclusively answer the question if such a connection exists, they show a practically relevant setting in which this correlation can be studied.

The discrepancy between the binning estimator and our estimator further suggests that either the information-theoretic compression we observe using our estimator is not geometric, or that there are insufficient samples to obtain reliable estimates from the binning estimator. This is in contrast with the work of Goldfeld et al. (2019), which showed that information-theoretic and geometric compression were linked in their networks with additive noise. We thus believe that a closer investigation of whether multiplicative noise induces geometric compression, and whether the induced compression improves generalization performance, are interesting questions for future research.

## ACKNOWLEDGEMENTS

The authors want to thank Michael Kamp, Simon Damm, Ziv Goldfeld, and Jihao Andreas Lin for valuable discussions about the work.

Asja Fischer acknowledges support by the Deutsche Forschungsgemeinschaft (DFG, German Research Foundation) under Germany's Excellence Strategy - EXC 2092 CASA - 390781972.

The Know-Center is funded within the Austrian COMET Program - Competence Centers for Excellent Technologies - under the auspices of the Austrian Federal Ministry of Climate Action, Environment, Energy, Mobility, Innovation and Technology, the Austrian Federal Ministry of Digital and Economic Affairs, and by the State of Styria. COMET is managed by the Austrian Research Promotion Agency FFG.

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

# A APPENDIX

## A.1 INFORMATION DROPOUT

One type of dropout with continuous noise is termed information dropout (Achille and Soatto, 2018). It is a technique that combines dropout noise sampled from a log-normal distribution $\epsilon \sim p_\epsilon = \log \mathcal{N}(0, \alpha_\theta^2(x))$, where $\alpha_\theta(x)$ is a learnable parameter dependent on the parameters $\theta$ of a network, and the introduction of a regularization term $KL(p_{z|x_i}||\prod_{i=1}^{|Z|} p_{z_i})$. This regularization term is based on an information bottleneck objective for training neural networks: Rewriting the information bottleneck Lagrangian and adding a disentanglement term (i.e., we want each element of representation $Z$ to be independent of the others) results in the aforementioned formula. Additionally, it is proposed to use as prior $p_z$, defined by the choice of activation function (ReLU or Softplus), a particular distribution whose validity is empirically verified. Such priors and selected dropout noise allow for deriving a closed form of KL-divergence, which makes it easy to directly track IP values while training.

In the following, we provide the closed form for computation of $I(X; Z)$ as proposed by Achille and Soatto (2018):

$$I(X; Z) = KL(p_{x,z}||p_z p_x) = \int p_{x,z}(x, z) \log \left( \frac{p_{x,z}(x, z)}{p_z(z) p_x(x)} \right) \mathrm{d}x \mathrm{d}z$$

$$= \int p_x(x) p_{z|x}(z) \log \left( \frac{p_x(x) p_{z|x}(z)}{p_z(z) p_x(x)} \right) \mathrm{d}x \mathrm{d}z = \int p_x(x) KL(p_{z|x}||p_z) \mathrm{d}x$$

$$= \mathbb{E}_x[KL(p_{z|x}||p_z)] .$$

Empirically we can approximate this as $I(X; Z) = \sum_{j=1}^{|S|} KL(p_{z|x^{(j)}}||p_z)$, where we sum over the dataset of size $|S|$ of samples of $X$.

First, we discuss ReLU neural networks. The prior distribution $p_z$ in this case consists is a mixture of two parts: and improper log-uniform distribution and a point mass at 0. Such prior is empirically valid for ReLU activations. First we restrict the derivation to the case when $f(X) \neq 0$ (which in turn means that $Z \neq 0$, since noise $\epsilon$ is log-normal and cannot be 0). In the following we will omit the subscript of probability density functions, when it is clear from its argument.

$$KL(p_{z|x^{(j)}}||p_z) = KL(p_{\log(z|x^{(j)})}||p_{\log(z)}) \tag{1}$$

$$= \int p(\log(z|x^{(j)})) \log \left( \frac{p(\log(z|x^{(j)}))}{p(\log(z))} \right) \mathrm{d}z$$

$$= \int p(\log(\epsilon) + \log(f(x^{(j)}))|x^{(j)}) \log \left( \frac{p(\log(\epsilon) + \log(f(x^{(j)}))|x^{(j)})}{c} \right) \mathrm{d}\epsilon \tag{2}$$

$$= \int p(\log(\epsilon)) \log(p(\log(\epsilon))) \mathrm{d}\epsilon - \int p(\log(\epsilon)) \log(c) \mathrm{d}\epsilon \tag{3}$$

$$= \int p(\log(\epsilon)) \log(p(\log(\epsilon))) \mathrm{d}\epsilon - \log(c) = -h(\log(\epsilon)) - \log(c) \tag{4}$$

$$= -(\log(\alpha(x^{(j)})) + \frac{1}{2}\log(2\pi e)) - \log(c) , \tag{5}$$

where equation 1 holds due to the invariance of the KL-divergence under parameter transformation with a strictly monotone function ($\log(\cdot)$); equation 2 holds since $\log(Z) = \log(\epsilon) + \log(f(X))$ and $p_{\log(z)} = c$ for the improper log-uniform distribution; equation 3 is taking into account that $p_{x+const} = p_x$, that $\log(f(x))|x^{(j)}$ is constant, and that $p_{\log(\epsilon)|x^{(j)}} = p_{\log(\epsilon)}$ because $\epsilon$ is independent of $X$; equation 4 uses that $\int p_{\log(\epsilon)} \mathrm{d}\epsilon = 1$; finally equation 5 holds because $\log(\epsilon)$ is normally distributed and its entropy can be computed in closed form.

Now we put $f(X) = 0$, and also get $Z = 0$. Then $p_{Z|X} = \delta_0$ (point mass or Dirac delta) and MI becomes:

$$KL(p_{z|x^{(j)}}||p_z) = \int p_{z|x}(z) \log \left( \frac{p_{z|x}(z)}{p_z(z)} \right) \mathrm{d}z = \int \delta_0 \log \left( \frac{\delta_0}{q\delta_0} \right) \mathrm{d}z = -\log(q) , \tag{6}$$

where $q$ is the weight of the point mass in the prior $p_z$.

Combination of equation 5 and equation 6 results in a computable $I(X; Z)$. As it can be seen, one has to correctly combine non-zero and zero values of $f(X)$ and also know the parameters of the prior $p_z$: constant $c$ and weight $q$. This makes it not practical for IP analysis.

If instead of ReLU the network has softplus activations, then the prior on the representations distribution is standard log-normal instead of log-uniform with delta Dirac. In this case the computation is very simple, since KL divergence between two log-normal distributions is computed as KL divergence between corresponding normal distributions:

$$KL(p_{z|x^{(j)}}||p_z) = \frac{1}{2\sigma^2}(\alpha^2(x^{(j)}) + \mu^2) - \frac{\log(\alpha(x^{(j)}))}{\sigma} - \frac{1}{2} \ , \tag{7}$$

where $\sigma^2 = 1$ and $\mu = 0$ are known parameters of the prior. Thus, softplus activations (equation 7) allows for direct computations of $I(X; Z)$.

## A.2 PROOF OF THEOREM 3.1

*Proof.* Using the chain rule of MI, we have

$$I(X; Z) = I(X; Z, B) - I(B; X|Z) = I(X; Z|B) + I(B; X) - I(B; X|Z)$$
$$\geq I(X; Z|B) - H(B)$$

where the inequality follows from dropping $I(B; X)$ since $B$ and $X$ are independent and the fact that $I(B; X|Z) \leq H(B)$. Having $B \in \{0, 1\}^{\hat{N}}$ as a discrete RV, it immediately follows that $H(B) \leq \hat{N} \log 2$. Now note that

$$I(X; Z|B) = \sum_{b \in \{0,1\}^{\hat{N}}} \mathbb{P}(B = b) I(X; Z|B = b).$$

Since the Bernoulli RVs are independent, positive probability mass is assigned to $b = (1, 1, \ldots, 1)$, i.e., to the case where all neurons are active. Evidently, when $b = (1, 1, \ldots, 1)$ it follows that $Z = f(X)$. Thus, with (Amjad and Geiger, 2020, Th. 1)

$$I(X; Z|B) \geq \mathbb{P}(b = (1, 1, \ldots, 1)) I(X; f(X)) = \infty$$

and $I(X; Z) = \infty$. $\qquad \square$

## A.3 PROOF OF THEOREM 3.2

*Proof.* If the binary dropout is such that nonzero probability is assigned to the dropout mask $b = (1, 1, \ldots 1)$, then the statement of the theorem follows as in the proof of the theorem 3.1.

Assume now that $B$ is such that zero mass is assigned to $b = (1, 1, \ldots, 1)$. To treat this case, we suppose that the distribution of $X$ has a portion with a continuous probability density function on a compact set and that the neural network has activation functions that are either bi-Lipschitz or continuously differentiable with a strictly positive derivative (following the requirements of Amjad and Geiger (2020, Th. 1)). Then, we obtain $I(X; f(X)) = \infty$ from (Amjad and Geiger, 2020, Th. 1) for almost all parameterizations of the neural network. Under this setting, $f_B(X)$ is again a neural network with activation functions that are either bi-Lipschitz or continuously differentiable with a strictly positive derivative. Assuming that $b$ is such that the input of the network is not completely disconnected from the considered layer, for this pattern we have $I(X; Z|B = b) = \infty$. Otherwise, we obviously have $I(X; Z|B = b) = 0$. The statement of the theorem follows from taking the expectation over all patterns $b$. $\qquad \square$

## A.4 PROOF OF THEOREM 3.3

*Proof.* W.l.o.g we first restrict our attention to the dimensions of representations $Z$ that are different from zero. Specifically, suppose that $Z = (Z_1, \ldots, Z_N)$ and that $B = (B_1, \ldots, B_N)$ with $B_i = 0$ if $Z_i = 0$ and $B_i = 1$ otherwise. Clearly, $B$ is a function of $Z$, hence $I(X; Z) = I(X; Z, B) = $

$I(B; X) + I(Z; X|B)$. Since $B$ is binary, we have that $I(X; B) \leq H(B) \leq n \log 2$. Let $Z_B = (Z_i|i: B_i = 1)$ denote the sub-vector of non-zero elements of $Z$, then

$$I(X; Z) \leq n \log 2 + \sum_b \mathbb{P}(B = b)I(Z_b; X)$$

where, if $B = b$, $I(Z_b; X) = I(Z; X|B = b)$ holds because constant (i.e., 0) RVs do not contribute to MI. Therefore, $I(X; Z)$ is finite iff $I(Z_b; X) = I(Z; X|B = b)$ is finite $B$-almost surely. We thus now fix an arbitrary $B = b$ and continue the proof for $Z = Z_b$.

We decompose MI into differential entropies as $I(X; Z) = h(Z) - h(Z|X)$. The differential entropy of the representations $h(Z)$ is upper-bounded by the entropy of a Gaussian RV with the same covariance matrix $\Sigma$ as the distribution of $Z = (Z_1, \ldots, Z_N)$, i.e., by $N/2 \log(2\pi) + 1/2 \log(\det(\Sigma)) + N/2$. From Hadamard's inequality and since $\Sigma$ is positive semidefinite it follows that $\det(\Sigma) \leq \prod_{i=1}^n \sigma_{ii}^2$, where $\sigma_{ii}^2$ are diagonal elements of the covariance matrix, i.e., $\sigma_{ii}^2 = Var[Z_i]$. This variance can be bounded from above. Specifically, since $X_i$ is bounded and $f(\cdot)$ is a composition of Lipschitz functions, $f(X)_i$ is bounded as well. Recalling that $\mathbb{E}[D_i(x)^2] \leq M$ holds $X$-almost surely, this yields

$$Var[Z_i] \leq \mathbb{E}_x[f(X)_i^2 D_i(X)^2] = \mathbb{E}_x[f(X)_i^2 \mathbb{E}_d[D_i(X)^2 \mid X]]$$
$$\leq M\mathbb{E}_x[f(X)_i^2] \leq M\|f(X)_i\|_\infty^2$$

It remains to show that the $h(Z|X) > -\infty$. Due to the conditional independence of $D_i$ and $D_j$ given $X$, for all $i \neq j$, the conditional differential entropy of $Z$ factorises in the sum of conditional differential entropy of its components, i.e., $h(Z|X) = \sum_{i=1}^N h(Z_i|X)$. We write this conditional entropy as an expectation over $X$ and obtain using (Cover and Thomas, 1991, Th. 9.6.4)

$$h(Z_i|X) = \mathbb{E}_x[h(Z_i|X = x)] = \mathbb{E}_x[h(D_i(x)|f(x)_i||X = x)]$$
$$= \mathbb{E}_x[h(D_i(x)|X = x)] + \mathbb{E}_x[\log(|f(X)_i|)]$$

by the formula of change of variables for differential entropy. Both terms are finite as per the assertion of the theorem. The first term is finite since we assumed that the differential entropy of $D_i(X)$ is essentially bounded, i.e., there exists a number $C < \infty$ such that $h(D_i(x)) \leq C$ $X$-almost surely. The second term is finite since we assumed that the conditional expectation $\mathbb{E}[\log(|f(X)|) \mid |f(X)| > 0]$ is finite in each of its elements, and since $Z_i \neq 0$ implies $|f(X)_i| > 0$. This completes the proof. $\square$

## A.5 Proof of Proposition 3.5

*Proof.* We assume w.l.o.g. that $f(\cdot)$ has a range with dimension $D = 1$, i.e., $f: \mathcal{X} \to \mathbb{R}$, where $\mathcal{X} \subseteq \mathbb{R}^n$ is the function domain. The proof can be straightforwardly extended to the several dimensions of $f(\cdot)$.

Since $f(\cdot)$ is constructed using a finitely-sized neural network with ReLU activation functions, it is piecewise affinely linear on a finite partition of the function domain. The fact that $\mathbb{E}[\log(|f(X)|) \mid |f(X)| > 0] < \infty$ follows then immediately from the fact that $\mathcal{X}$, and thus $|f(\mathcal{X})|$, is bounded.

To investigate whether $\mathbb{E}[\log(|f(X)|) \mid |f(X)| > 0] > -\infty$, split domain $\mathcal{X}$ in the following partitions:

1. $\mathcal{X}_0 = f^{-1}(\{0\})$ denotes the element of the partition on which $f(X)$ vanishes;

2. $\{\mathcal{X}_i^c\}_{i=1,\ldots,\ell}$ denotes elements of the partition of $\mathcal{X}$ on which $f(X) = c_i$, i.e., on which $f(\cdot)$ is constant;

3. $\mathcal{X}^a = \bigcup_{i=1}^m \mathcal{X}_i^a$ denotes the union of the all other sets $\{\mathcal{X}_i^a\}_{i=1,\ldots,m}$ of the partition, where $f(\cdot)$ is not constant.

For the last subset, define the function $\tilde{f}: \mathcal{X}^a \to \mathbb{R}^n$ via $\tilde{f}(x) = (|f(x)|, x_2, x_3, \ldots, x_n)$. Note that $\tilde{f}(\cdot)$ is piecewise bijective, hence $\tilde{W} = \tilde{f}(X)$ has a probability density function that is obtained

from the change of variables formula:

$$p_{\tilde{w}}(\tilde{w}) = \sum_{x \in \tilde{f}^{-1}(\tilde{w})} \frac{p_x(x)}{|\det(J_{\tilde{f}}(x))|}$$

where $J_{\tilde{f}}(x) = \left[\frac{\partial \tilde{f}_i}{\partial x_j}(x)\right]$ is the Jacobian matrix of $\tilde{f}(\cdot)$, with $\tilde{f}_1(x) = |f(x)|$ and $\tilde{f}_j(x) = x_j$ for all $j \geq 2$. It follows that Jacobian matrix is diagonal and has determinant $|\frac{\partial f}{\partial x_1}(x)|$. The density $p_{w|\mathcal{X}^a}$ of the conditional random variable $W = |f(X)| \mid X \in \mathcal{X}^a$ can be then obtained by marginalization from $p_{\tilde{w}}$:

$$p_{w|\mathcal{X}^a}(w) = \int p_{\tilde{w}}(w, x_2^n) \mathrm{d}x_2^n = \int \sum_{x \in \tilde{f}^{-1}(w, x_2^n)} \frac{p_x(x)}{|\frac{\partial f}{\partial x_1}(x)|} \mathrm{d}x_2^n \tag{8}$$

where $x_2^n = (x_2, \ldots, x_n)$ and where we perform an $(n-1)$-fold integral.

Thus, by the Lebesgue decomposition, the distribution of $W = |f(X)|$ can be split into an absolutely continuous component with a probability density function $p_{w|\mathcal{X}^a}$ and a discrete component with finitely many mass points, for which we have $\mathbb{P}(W = c_i) = \int_{\mathcal{X}_i^c} p_x(x) \mathrm{d}x =: p_x(\mathcal{X}_i^c)$. By the law of unconscious statistician, we then obtain

$$\mathbb{E}[\log(|f(X)|) \mid |f(X)| > 0]$$
$$= \mathbb{E}[\log(W) \mid W > 0]$$
$$= \sum_{i=1}^{\ell} p_x(\mathcal{X}_i^c) \log|c_i| + p_x(\mathcal{X}^a) \int_0^\infty \log(w) p_{w|\mathcal{X}^a}(w) \mathrm{d}w$$
$$= \sum_{i=1}^{\ell} p_x(\mathcal{X}_i^c) \log|c_i| + p_x(\mathcal{X}^a) \underbrace{\int_0^\epsilon \log(w) p_{w|\mathcal{X}^a}(w) \mathrm{d}w}_{I_1} + p_x(\mathcal{X}^a) \underbrace{\int_\epsilon^\infty \log(w) p_{w|\mathcal{X}^a}(w) \mathrm{d}w}_{I_2}$$

where in the last line we split the integral at a fixed $\epsilon \ll 1$. Clearly, the first sum is finite since $c_i > 0$ for all $i$. For the remaining summands involving integrals, suppose for now that $p_{w|\mathcal{X}^a}(w) \leq C < \infty$. Then,

$$I_1 = \int_0^\epsilon p_w \log(w) dw \geq \int_0^\epsilon C \log(w) dw = C(\epsilon \log(\epsilon) - \epsilon) > -\infty$$
$$I_2 = \int_\epsilon^\infty p_w \log(w) dw \geq \int_\epsilon^\infty p_w \left(1 - \frac{1}{w}\right) dw \geq \int_\epsilon^\infty p_w \left(1 - \frac{1}{\epsilon}\right) dw \geq 1 - \frac{1}{\epsilon} > -\infty.$$

We thus remain to show that $p_{w|\mathcal{X}^a}(w) \leq C$ for $w \in [0, \epsilon]$. To this end, we revisit equation 8 and note that the integral is finite if i) $p_x$ is bounded, ii) the integration is over a bounded set, and iii) $|\frac{\partial f}{\partial x_1}(x)| \geq \epsilon_1 > 0$. Conditions i) and ii) are ensured by the assertion of the lemma. It remains to show that condition iii) holds.

Note that in contrast to using $\tilde{f}(x) = (|f(x)|, x_2, x_3, \ldots, x_n)$, the same $p_{w|\mathcal{X}^a}(w)$ can also be obtained by using the piecewise bijective function $\tilde{f}(x) = (x_1, |f(x)|, x_3, \ldots, x_n)$, etc. Hence, $p_{w|\mathcal{X}^a}(w) \leq C$ if the partial derivative of $f$ is bounded from below for at least one dimension, i.e., if there exists an $i$ such that $|\frac{\partial f}{\partial x_1}(x)| \geq \epsilon_1$. Since we have

$$\|\nabla_x f(x)\|_1 = \sum_{i=1}^n \left|\frac{\partial f}{\partial x_i}(x)\right|$$

this is equivalent to requiring that the $L_1$ norm of the gradient is bounded from below. Indeed, remember that $f$ is piecewise affinely linear with finitely many pieces, and its restriction to $\mathcal{X}^a$ is non-constant. On its restriction to $\mathcal{X}^a$ we thus have $\nabla_x f(x) = g_i > 0$ for all $x \in \mathcal{X}^a$ and some $i \in \{1, \ldots, m\}$. Hence, we can find an $\epsilon_1$ such that $\min_i g_i \geq n \cdot \epsilon_1 > 0$, which implies that there exists an $i$ for which $|\frac{\partial f}{\partial x_i}(x)| \geq \epsilon_1$ for all $x \in \mathcal{X}^a$. This completes the proof. $\qquad \square$

### A.6 ESTIMATION OF MI UNDER GAUSSIAN DROPOUT

In the Algorithm 1 we describe how the estimation of $I(X; Z)$ with $Z$ being a representation under Gaussian dropout can be done. This is the way we estimated MI for our experiments, but any other estimator can be used in this setup.

---

**Algorithm 1** Estimation of MI under Gaussian dropout

---

**Require:** GMM-MEANS, $\sigma$, nonoise-reprs      ▷ Amount of Gaussians in GM for approximation;
     noise variance; no noise representations
     $reprs \leftarrow []$                       ▷ Generate noisy samples with corresponding variance
     **for all** $nr$ in nonoise-reprs **do**
         **for** $i \leftarrow 1, n$ **do**
             $\epsilon \leftarrow noise_p$
             $reprs \leftarrow reprs + nr * \epsilon$
         **end for**
     **end for**
     points $\leftarrow$ nonoise-reprs[: GMM-MEANS]      ▷ Create a GMM on restricted amount of points for
     faster computation
     $d \leftarrow []$
     **for all** $p$ in $points$ **do**
         $d \leftarrow d + \text{Gaussian}(p, \sigma * |p|)$
     **end for**
     gmm $\leftarrow$ MixtureModel($d$)
     $lp \leftarrow []$                        ▷ Get estimates of log-probabilities from GMM for noisy samples
     **for all** $r$ in reprs **do**
         $lp \leftarrow lp + \text{gmm.log\_probability}(r)$
     **end for**
     $h(z) \leftarrow \text{mean}(lp)$
     $h(z|x) \leftarrow 0$                 ▷ Compute conditional entropy using closed form formula
     **for** $i \leftarrow 1, \dim(\text{reprs}[0])$ **do**             ▷ For each dimension of the representation
         $h(z|x) \leftarrow h(z|x) + \text{mean}(ln(\sqrt{2\pi e}\sigma|\text{nonoise-reprs}[:, i]|))$      ▷ Use no noise representations
     here, each dimension separately
     **end for**
     $I(x, z) \leftarrow h(z) - h(z|x)$                      ▷ Obtain final estimate for the MI

---

### A.7 EVALUATION OF ESTIMATOR

Fig. 8 shows upper bounds and estimation of $h(Z)$ with a higher noise than in the Fig. 3. Larger noise increases the gap between the Gaussian entropy based upper bound and the mixture based estimation as expected.

In Fig. 9 we see convergence of the MC estimate for $h(Z|X)$ under larger noise.

As expected larger noise variance results in smaller MI values (Fig. 10), while the trend observed when changing dimensionality stays the same.

### A.8 INFORMATION PLANE ANALYSIS

Note, that in the experiments we analyze IPs on the training samples and test samples separately. In order to obtain a valid sample of hidden representations for the MI estimation during inference, we apply MC-Dropout, as opposed to the usual way of performing inference with dropout being turned off. According to Srivastava et al. (2014) this is the theoretically sound way to obtain predictions, while turning off dropout and re-scaling weights results in an approximationthat allows for faster computation.

In Fig. 11, Fig.12, and Fig. 13 we provide IPs built on the test set of the corresponding datasets (MNIST, MNIST, and CIFAR10).

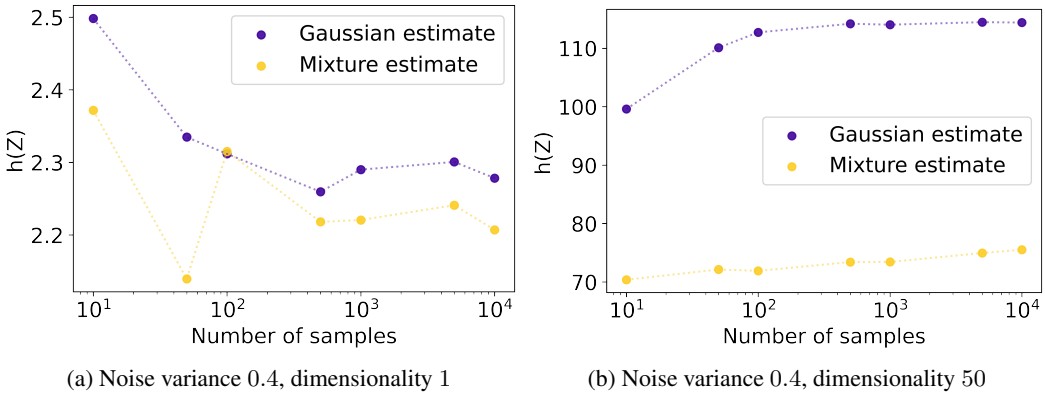

(a) Noise variance 0.4, dimensionality 1      (b) Noise variance 0.4, dimensionality 50

Figure 8: Entropy of the hidden representation. It can be seen that with growing dimensionality the Gaussian upper bound becomes very loose, compared to the Gaussian mixture estimation.

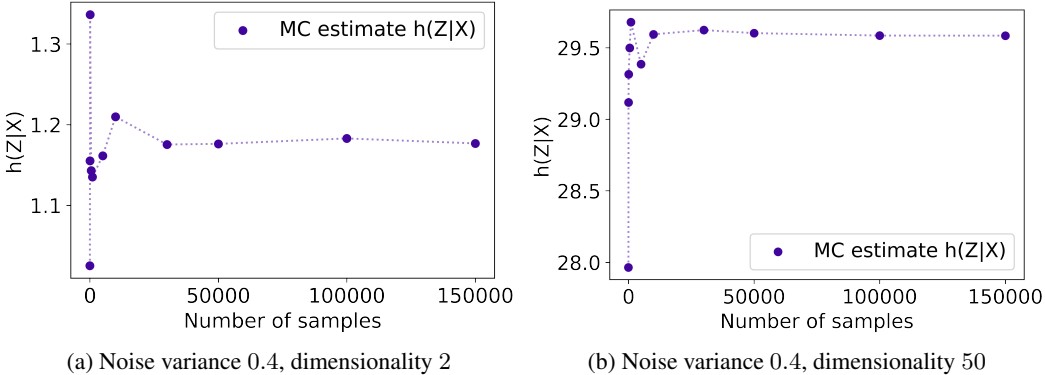

(a) Noise variance 0.4, dimensionality 2      (b) Noise variance 0.4, dimensionality 50

Figure 9: Conditional entropy of the hidden representation. Independent of the dimensionality the MC estimation of $h(Z|X)$ stabilizes with increasing amount of samples.

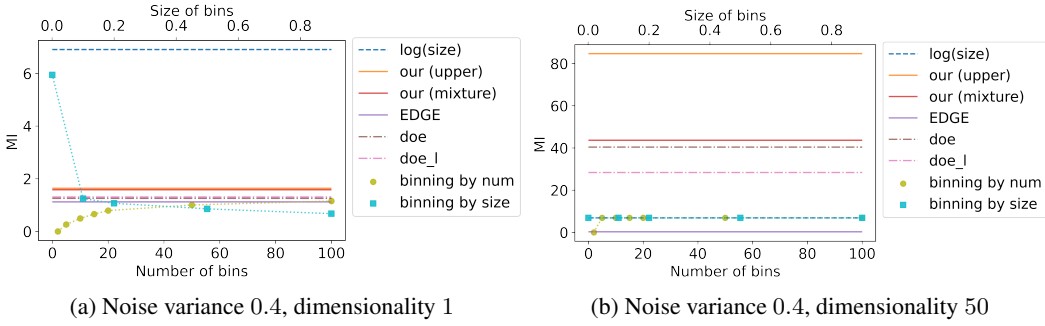

(a) Noise variance 0.4, dimensionality 1      (b) Noise variance 0.4, dimensionality 50

Figure 10: Comparison of various approaches to MI estimation for the setup of the multiplicative Gaussian noise.

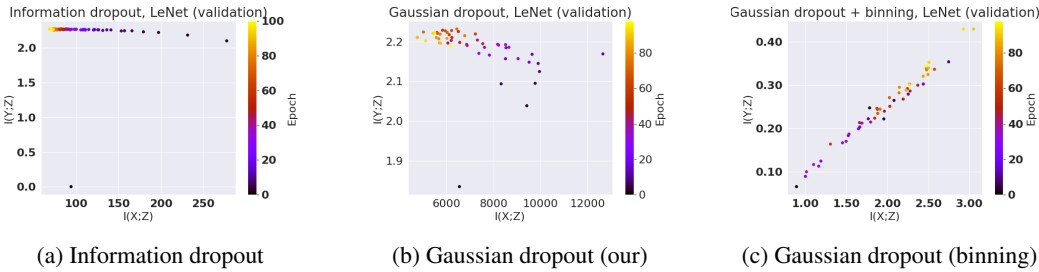

(a) Information dropout      (b) Gaussian dropout (our)      (c) Gaussian dropout (binning)

Figure 11: Compared to the IP analysis based on binning (discrete) estimation of MI, the IP based on our approach shows compression as well for Gaussian as for information dropout.

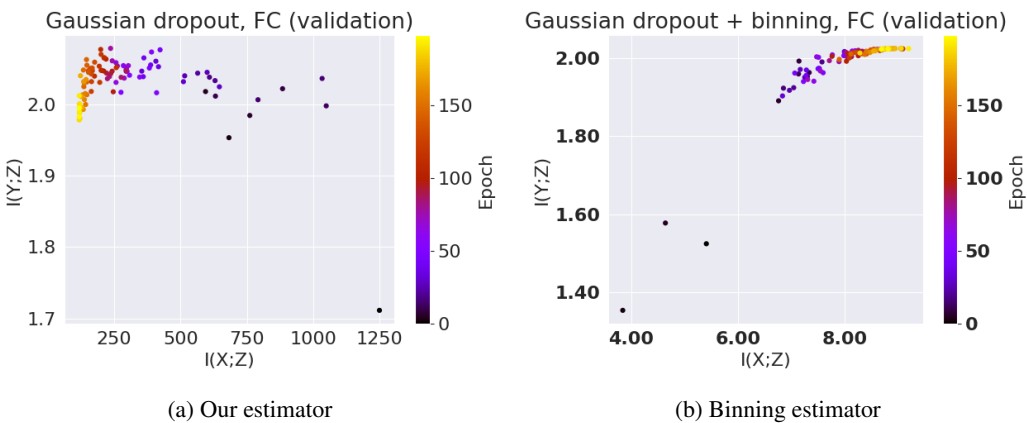

(a) Our estimator      (b) Binning estimator

Figure 12: Compared to the IP analysis based on binning (discrete) estimation of MI, the IP based on our approach shows compression.

In the Fig. 14 we provide additional IPs for the binning estimator with varying amount of bins used for MI estimation. We report the results for the fully-connected network trained on MNIST with Gaussian dropout variance $0.2$.

In the Fig. 15 we show the IPs obtained for the same fully-connected network trained on MNIST with the variance of the Gaussian dropout set to $0.4$.

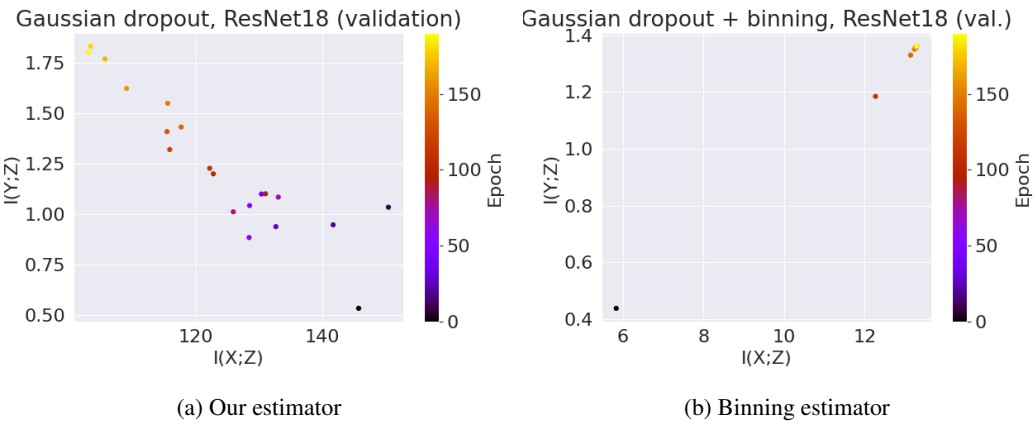

(a) Our estimator

(b) Binning estimator

Figure 13: Comparing to the discrete estimation of MI for an IP analysis our approach clearly shows compression.

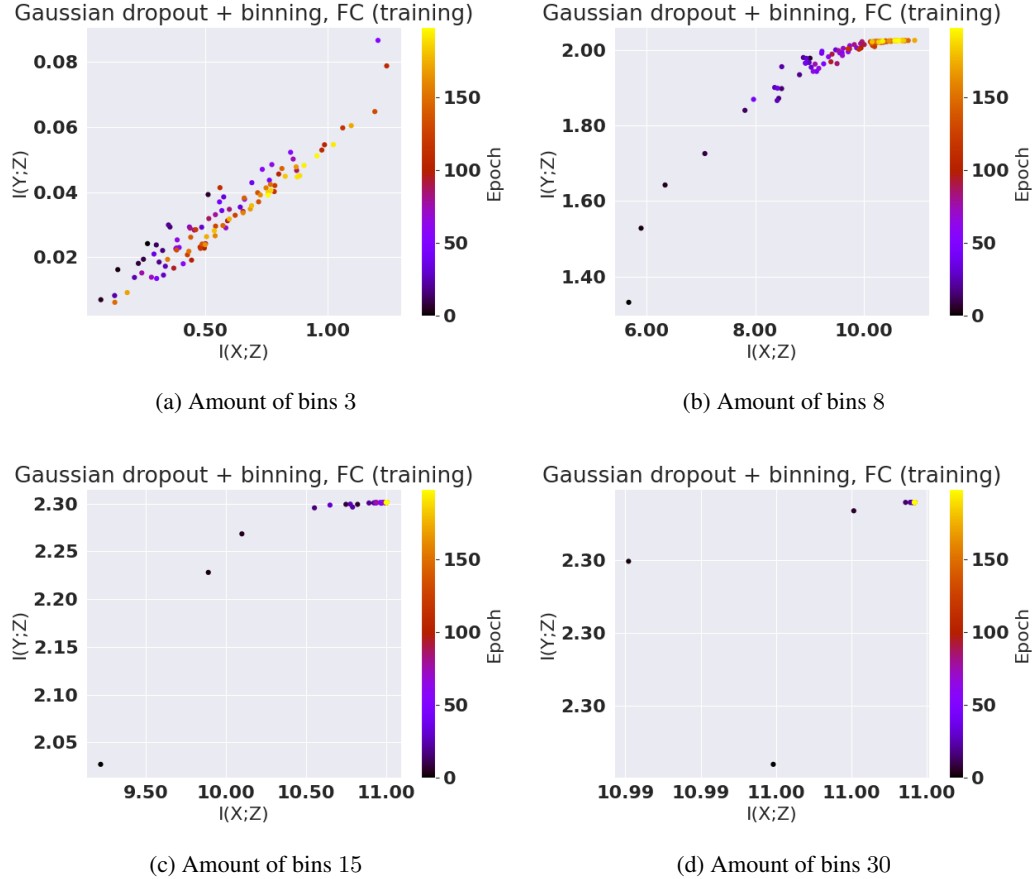

(a) Amount of bins 3

(b) Amount of bins 8

(c) Amount of bins 15

(d) Amount of bins 30

Figure 14: Larger amount of bins used for estimation of MI leads to collapse of the IP into one point corresponding to the amount of samples available. All the smaller amount of bins demonstrates no compression in terms of $I(X;Z)$.

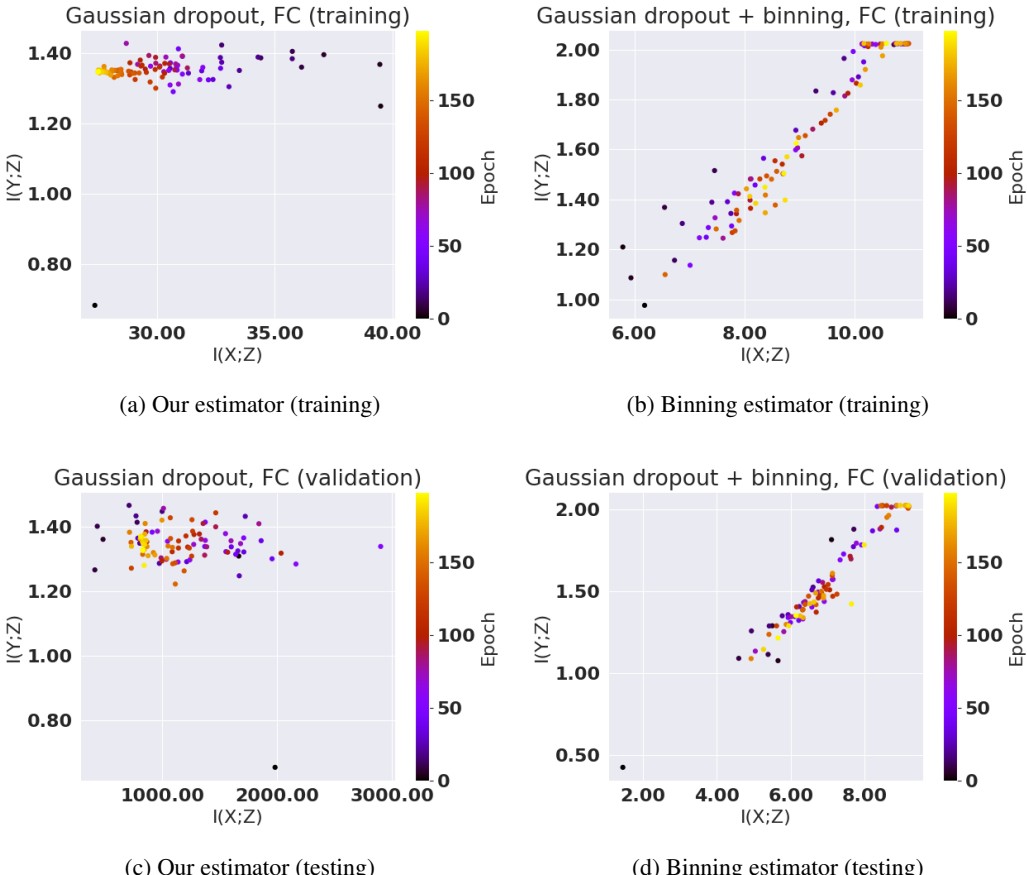

(a) Our estimator (training)

(b) Binning estimator (training)

(c) Our estimator (testing)

(d) Binning estimator (testing)

Figure 15: Same as for the variance of the dropout 0.2 we observe compression when measured with our estimator compared to no compression with binning.

