# OpenReview forum: "Information Plane Analysis for Dropout Neural Networks"
_ICLR.cc/2023/Conference — ICLR 2023 poster_

### Official Review · Reviewer_iYGf · 2022-10-20

**Confidence:** 3
**Correctness:** 3
**Technical Novelty And Significance:** 2
**Empirical Novelty And Significance:** 3
**Recommendation:** 5

**Clarity, Quality, Novelty And Reproducibility:**

# Clarity
* This is a minor point. The colors of lines in Figure 4 are similar and hard to recognize.

# Novelty
* The idea of using dropout is interesting, but it is the straightforward extension of the idea of stochastic neural networks. As far as I understood the technique is not quite novel.

**Strength And Weaknesses:**

# Strength
* Compared to stochastic neural networks, using the dropout is more practical and realistic.
* The estimation procedure of MI using the dropout seems easy and worked well compared to binning approaches.

# Weakness
* Compared to binning approaches, this method only applies to the dropout models with continuous noise. There are many networks that do not use dropout and continuous noise.
* When using information dropout, its noise is learned simultaneously. This means the estimator of MI changes during the training. Thus, we use different MI estimators for different epochs when plotting the IP. So it is hard to understand the obtained IP curves or meaningless. Is my understanding correct?
* I wonder how the dropout hyperparameters affect the estimation of MI and the IP. For example. The number of dropout layers or the magnitude of the noise should have a large impact on them, but there is no discussion and no numerical experiments about them.
* Why I(Y; Z) are so different between dropout and binning methods, as shown in Figures 1, 5, 11, 12? As far as I understand, I(Y; Z) is estimated by subtracting H(Y) from the cross entropy as mentioned in Sec 4. This is the same procedure for the dropout and binning approaches.

**Summary Of The Paper:**

This paper address the ill-posed problem of IB of deterministic neural networks. Motivated by the success of stochastic neural networks, the authors proposed using the dropout technique to define and estimate the mutual information in neural networks. The authors showed that MI is bounded when the dropout uses continuous noise. Then they conducted numerical experiments of MI and studied the IP of neural network models.

**Summary Of The Review:**

This paper proposed elegant way of defining and estimating MI and IP for networks that uses dropout. However, as far as I understood, the technique is not so novel and numerical experiments are limited to show the usefulness and understand the IP in dropout models.

---

> ### Author Response · Authors · 2022-11-15
> **Reply**
>
> We thank the reviewer for the valuable feedback.
>
> ## Limited applicability compared to binning
> We want to emphasize again that we do not propose continuous dropout as an estimator for MI – see general comment to all the reviewers. If a NN does not contain stochasticity, then it does not make sense to estimate MI in the first place (and discrete noise is not sufficient as we showed also in our Theorems 3.1 and 3.2 for binary dropout). The MI is infinite in this setting, and estimation results will differ only due to the errors induced by discrete approximation
> Binning is an estimation technique that can be used for any NN (including deterministic NNs or NNs with continuous dropout), but it regularly suffers from the curse of dimensionality and/or the fact that often insufficient data is available for estimation. Our estimators utilize knowledge of the network architecture and can be assumed to be more efficient in their need for data.
>
> ## MI in networks with information dropout
> In the case of information dropout the MI is computed in a closed form (which depends on the noise/variance), using the special types of distributions and variational computations (that are recaptured in the Appendix from the original paper of Achille). So there is no estimator used except for Monte Carlo sampling over the input space. Therefore, the changing noise does not pose a problem.
>
> ## Effect of hyperparameters
> We thank the reviewer for the idea to try out different noise levels of the dropout. We conjecture that the increase of the noise level (as it can be seen in the experiments in Section 4 and appendix) will reduce the estimated MI values as expected, but it can also result in different behavior reflected in the IP.
> Indeed, we agree that studying the effect of continuous dropout at multiple layers is interesting. Since this requires adapting our estimation techniques accordingly, such an analysis is within the scope of future work.
> ## Difference in I(Z;Y) estimates
> The estimation of this value can be done in multiple ways. The method mentioned by the reviewer was used for information dropout experiments (since in those experiments representations were not saved for further computations). For the experiments with Gaussian dropout we employed either EDGE estimator or binning itself (when IP was built for the binning estimator). We clarified this in Section 5.
>
> ## Clarity
> We improve the readability of Figure4.
>
> ## Novelty of the proposed technique
> We argue that the proposed analysis is novel: while neither binary nor continuous dropout networks are novel, our contribution is a theoretical analysis of whether these networks can be studied using the information plane, i.e., whether MI between input and latent representation is finite or infinite. Our novel results show that using binary dropout does not result in finite MI, while continuous does. We further do a preliminary analysis of the possibilities that estimation of MI in dropout networks can give, in particular, how the IP might differ if the networks are in this setup. Our main focus is to enable possible further insights from the (theoretically grounded) MI estimation in neural networks.

---

### Official Review · Reviewer_e4St · 2022-10-20

**Confidence:** 4
**Correctness:** 4
**Technical Novelty And Significance:** 3
**Empirical Novelty And Significance:** 3
**Recommendation:** 8

**Clarity, Quality, Novelty And Reproducibility:**

Novelty
- Both the idea and realization of using stochasticity hidden in dropout layers to estimate MI are novel. The MI estimates seem to be accurate enough to provide a meaningful information plane analysis of the model.
- Related work is cited and analyzed adequately.

Quality
- The technical and experimental results seem to be well-executed to the best of my assessment. I appreciate namely the detailed theoretical analysis.

Clarity And Reproducibility

- The paper is written comprehensibly and its structure is good.
1. Nevertheless, the description and computation of the proposed MI estimate is "hidden" in the paragraphs.
 I recommend the authors to summarize the computation into one comprehensible expression (or in a box with pseudocode). This will help readers to quickly understand and reproduce it.
2. The colors in Figure 4 and Fig. 10 are not well chosen. It impossible to distinguish the individual shades of orange/red. That makes the graphs incomprehensible.
Please, change the colors and support the comprehensibility by dashed lines.
3. Figures and their labels are too small (compared to other text) and therefore hardly comprehensible without high magnification on the screen (e.g., Figure 1, 5, 6, 7).
4. It is not clear what "doe" and "doe_l" stand for in Fig. 4 and Fig. 10. Please add an explanation of this notation.



**Strength And Weaknesses:**

Plus

 A solid contribution to the area of information-theoretic analysis of neural networks.
-  The idea of utilizing stochasticity induced by dropout layers to estimate MI (between input and internal model representation) is novel and interesting.
-  The authors provide a thorough theoretical analysis of MI estimation in NNs and its limitations for both binary dropout and dropout with continuous noise.
-  The proposed monte-carlo MI estimate seems promising (based on the experiments).


Minus

I see the main limitation of the paper in relatively narrow area of ​​application. The method is restricted to dropout networks with continuous noise.
The authors provide a proof that the principle cannot be extended to (the widely used) binary dropout (or to NNs without dropout).
The concurrent technique of (Goldfeld et al. 2019) seems to offer wider possibilities of use despite its other disadvantages (namely the need to alter the internal representation of the model by noise).



**Summary Of The Paper:**

The goal of the paper is to obtain a sufficiently accurate estimate of MI (between input and representation) for dropout neural networks  and to use it to confirm the information bottleneck hypothesis for this NN model.

Contributions:
- The authors propose a monte-carlo based estimate of MI in Gaussian dropout networks that is relatively accurate and well-supported by the performed theoretical analysis.
- The authors use their MI estimate to provide information plane analyses for several NN models with Gaussian and information dropout (LeNet and MLP on MNIST, ResNet on CIFAR10)



**Summary Of The Review:**

A solid contribution without greater flaws except limited area of application.

---

> ### Author Response · Authors · 2022-11-15
> **Reply**
>
> We thank the reviewer for the positive feedback and appreciation of the proposed idea.
>
> ## Narrow area of application
> We understand the concern of the reviewer regarding the limitation of the results to dropout networks. However, we note that multiplicative noise could just be inserted into deterministic networks as it is proposed for additive noise by Goldfield et al., sharing the same disadvantage of altering the network of interest. Moreover, even if binary dropout is more frequently used in practice, we believe it can easily be replace by Gaussian dropout in practice without sacrificing performance (already the original paper introducing dropout (Srivastava, 2014) has a discussion about continuous dropout and concludes that it might be even more beneficial than binary.)
>
> ## Pseudocode
> We thank the reviewer for the idea to put the estimation algorithm for MI in a separately clearly written part. We have done this (see appendix A4)  and also we plan on open sourcing the code with a separate library for performing these estimations.
>
> ## Readability of the figures
> We improved the readability of the plots 4 and 10, as well as text in other plots in the revised version and added the description of the baselines in plots 4 and 10 (from the paper by McAlister).

---

### Official Review · Reviewer_DSDh · 2022-10-24

**Confidence:** 4
**Correctness:** 4
**Technical Novelty And Significance:** 4
**Empirical Novelty And Significance:** 4
**Recommendation:** 8

**Clarity, Quality, Novelty And Reproducibility:**

The paper is clearly written.  There are a few small grammatical errors and if the the authors have the energy they might consider rereading the text once more to eliminate these.  The paper is a high quality paper (the proofs although in a way straightforward requires a sophisticated understanding of probability which is rare).  The work is novel and the proofs and empirical results reproducible.

**Strength And Weaknesses:**

The paper resolves a long standing problem with using information theory to study learning in deep neural networks and particularly computing mutual informations.  This is a problem that has received an enormous amount of interest.  The proposed solution is both technically sophisticated (in terms of proofs) and elegant (in terms of implementation).

My very personal question as a naturally sceptical person is whether all the effort put into studying the information plane is worth all the effort. I'm prepared to accept that there are enough people interested that this is worth publishing, but the results section did not convince me that there is a lot of understanding to be gained.  I accept though that the authors have said that a full analysis will end up elsewhere so I guess I need to wait.  I also have a rather practical question that these measure are intrinsically super unstable when the mutual information becomes unbounded in a deterministic network.  This gives me an uncomfortable feeling as this does not reflect the generalisation performance of networks with and without dropout (or with continuous versus discrete dropout).  It seems to me that the true solution is to find a more meaningful picture than the information measure where networks that perform similarly don't have such different behaviour.  However, I realise this is just me being picky and I'm not really expecting the authors to address my scepticism,

**Summary Of The Paper:**

The paper analyses the computation of mutual information in DNN with dropout.  It shows that mutual information for discrete dropout is infinite, but for continuous drop it is a well defined finite quantity.  The paper shows how this can be estimated using Monte Carlo techniques and empirically shows that this quantity converges.  The paper finishes with an analysis of training a neural network with dropout in the information plane.

**Summary Of The Review:**

This is a well written paper addressing a problem that has received considerable attention in the research community.  It provides a sensible way of making sense of the Information Plane.  It is mathematically sophisticated, but leads to a practical application.  In my view this is of sufficient interest to warrant publication.

---

> ### Author Response · Authors · 2022-11-15
> **Reply**
>
> We thank the reviewer for the encouraging and supportive feedback! We are very happy that our idea is highly appreciated. We reread the paper and fix grammatical errors and typos up to our knowledge and rewrote some parts of the paper to improve clarity.
>
> ## Skepticism about information plane analysis
> We do confirm the skepticism of the reviewer: whether or not information-theoretic compression is correlated with improved generalization is the main question connected to and the most prominent justification for information plane analyses. Such a connection can only be tested for stochastic (in some way) NNs for which MI is finite and therefore measurable. Thus, most of the previous works did not contribute to answering that question. While our own work also does not answer the question (and the experiments section includes only preliminary experiments on the topic), it shows a practically relevant setting in which this correlation can be studied (by showing that MI is finite for networks with continuous dropout). Also, in our experiments we show that the binning estimator behaves very differently from our estimator, which indicates that either the information-theoretic compression is NOT geometric, or that there are insufficient samples to obtain reliable estimates from the binning estimator. We make this more clear in the revised manuscript.
> We believe that based on this widely used stochasticity further investigation of information-theoretic perspective can be made, allowing to shed light to the phenomena that are still hard to understand using only the loss curves.
> With this being said, information-theoretic compression may still be neither necessary nor sufficient for good generalization, which is why we may not see it in deterministic NNs. However, it may be that there is a correlation between the two, and this deserves to be studied in those settings in which information-theoretic compression is actually possible.
> We also find it very interesting to analyze a possible connection between geometric and information-theoretic compression. Since the majority of the previous work did not use appropriate estimators to study information-theoretic compression and since the work of Goldfeld et al. showed that information-theoretic and geometric compression were linked in their networks with additive noise, our results indicating that geometric and information-theoretic effects may be different for networks with multiplicative noise seems interesting to us.

---

### Official Review · Reviewer_A9Ut · 2022-11-03

**Confidence:** 3
**Clarity, Quality, Novelty And Reproducibility:** The paper is relatively easy to read,…
**Correctness:** 3
**Technical Novelty And Significance:** 2
**Empirical Novelty And Significance:** 1
**Recommendation:** 3

**Strength And Weaknesses:**

STRENGTHS

The interpretation of neural networks with the information plane approach is a relevant problem which has garnered some attention since it was proposed in 2017.

The authors' idea of using dropout for MI estimation is valid, although not necessarily novel.


WEAKNESSES

The main weakness of the paper is its limited novelty. The negative results presented in Section 3 are straightforward and the only new results of this paper follow from the restriction to Gaussian dropout and estimating entropies with MC methods. The observed results concerning the compression phase in the last section are not convincing (the curves resemble straight lines). The authors also only compare this result to the binning estimator which is known to result is spurious effects (see e.g. the works of Saxe or Gabrie).


**Summary Of The Paper:**

This paper puts forward a method of mutual information estimation within neural networks where the stochasticity of observations is ensured by Gaussian dropout. The authors then use MC methods to estimate entropies and conditional entropies. The proposed approach is compared to MI estimation based on binning and the authors claim to observe the compression phase postulated by Tishby et al., which was not observed with the binning estimator.


**Summary Of The Review:**

This paper takes an interesting approach to the analysis of NN with the information plane, but the presented results seem neither significant nor novel enough to warrant publication.

---

> ### Author Response · Authors · 2022-11-15
> **Reply**
>
> We thank the reviewer for their feedback.
>
> ## Limited Novelty
> We believe that the negative result proving that binary dropout does not prevent I(X;Z) from getting infinite – even if not difficult to derive – is novel and interesting. More generally, it suggests that the stochasticity has to be of a particular form to prevent mutual information from becoming infinite. While this may appear intuitive in hindsight, to the best of our knowledge we are the first to make this explicit, and thus complement the literature (e.g., Amjad, et al. 2019; Goldfeld, et al. 2019) claiming that stochastic networks may be candidates for a valid IP analysis. Furthermore, we give a theoretical justification for estimating the MI in dropout networks with continuous noise. (In the revised version we also added an analogous statement for stochastic networks with additive noise as used by Goldfeld et al. 2019). Without showing that the MI indeed is finite it is not guaranteed that an IP analysis is meaningful. Therefore, we believe that our theoretical analysis is of high importance on its own and see it as the main contribution of our paper.
>
> ## Comparison with Binning Estimators
> It is true that binning estimators are known to produce spurious results. Indeed, it has been shown that for deterministic networks MI is infinite, hence an estimation of MI is meaningless. Our aim was to show that, despite the fact that MI is finite and thus its estimation is well-posed for networks with continuous dropout, the binning estimator yields qualitatively different results than our estimator, which utilizes knowledge about the process of how Z is derived from X. Whether this is because the picture provided by binning estimators is dominated by geometric effects, or by the fact that insufficient data is available for estimation, is not clear yet and will be the subject of future study. In this sense, our work parallels the ones of Goldfeld et al. or Gabrie et al., which also studied binning estimators in settings with finite MI.
>
> ## Straight lines in the IP are not convincing
> To properly respond to this critique, we would appreciate the respective figure identifiers that the reviewer finds unconvincing. That said, we agree that straight lines in the information plane are uncommon when compared to the existing literature. We claim, however, that this is exactly what sets our analysis apart from it, and that this is in itself an interesting contribution. Note also that the direction of straight lines is relevant: vertical lines indicate no compression, diagonal lines indicate either an increase of I(X;Z) or compression, depending on whether they point to the right or to the left, respectively. Note further that the trajectories in Fig. 5a and Fig. 7 exhibit less straightforward behavior.

---

### Comment · Area_Chair_fLG6 · 2022-11-15
**Please engage before the author-reviewer discussion closes**

Dear authors and reviewers,

The first phase of the discussion period is about to close on November 18.

For authors, please make sure to submit your rebuttal by the deadline. Leave some time for the reviewers to read it and respond while you are still allowed to further engage with them. Interactions between authors and reviewers are very important for the quality of the review process, so please make sure to engage.

For reviewers, please try to acknowledge and respond to the authors' rebuttal while the discussion period is still open for them to further interact with you.

Thank you for your participation in the review process!

Best,
The AC

---

### Author Response · Authors · 2022-11-15
**Note to all reviewers**

We thank all the reviewers for the feedback and engagement.

We uploaded a revised version of our manuscript, in which we - encouraged by suggestions of the reviewers -  made the following changes:
- We revised the entire document (especially the introduction and Section 4). We show that networks with continuous dropout have finite mutual information between input and representation and we propose practical estimators for it. The contribution of our work is hence not a novel method, but a theoretical result about the applicability of the information plane analysis for networks with continuous dropout. Our empirical results show that using our theoretically sound MI estimation results in an information plane analysis with different properties than in previous works.
- We added a statement about the MI in stochastic networks with additive noise being finite as well.
- We added IPs based on binning estimators with different binning sizes in the appendix A4. The results show that only a small range of binning setups results in a readable IP and none of them demonstrates compression.
- We added IPs for Gaussian dropout networks with varying amounts of noise in the appendix A4. The results are similar to the previous.
- We improved the presentation of Figure 4 and 10 and enlarged the font size in (almost all) the plots. (The final editions will be made in the next few days.)

---

> ### Author Response · Authors · 2022-12-12
> **Update**
>
> Dear reviewers, dear AC,
>
> Since the time for discussion is running out, this is just a short reminder for the answers and revisions.
> I hope our replies helped understanding and revised version satisfies the comments.
>
> Best regards,

---

### Decision · Program_Chairs · 2023-01-20

**Decision:**

Accept: poster

**Justification For Why Not Higher Score:**

Given the initial concerns of two reviewers, I do not recommend a higher score.

**Justification For Why Not Lower Score:**

Valuable theoretical contribution. Interesting experimental obervations. Technically solid.

**Metareview: Summary, Strengths And Weaknesses:**

The paper has received mixed reviews, with two reviewers strongly recommending acceptance (8-8) and two reviewers recommending rejection (3-5). The author-reviewer discussion was productive and the authors addressed many of the reviewers' concerns, although they have not expressed whether they are satisfied with the authors' responses.

The paper presents a theoretical result that, in my opinion, constitutes a valuable stepping stone towards the understanding of the training dynamics of neural networks. The paper studies mutual information estimates in neural networks with dropout noise and finds that binary dropout does not prevent MI from being infinite. The analysis follows with an information plane analysis of neural networks with continuous dropout, leading to the (supposed) observation of compression during training, as postulated by Tishby et al. (2017).

As an area chair, I believe that the contribution of the paper is interesting and novel. It is technically well done and the experimental results reveal interesting insights. Concerns about the limited novelty are not strongly justified. I recommend accepting the paper for publication.

**Note From Pc:**

if the above contains the word "oral" or "spotlight" please see: "oral" presentation means -> notable-top-5% and "spotlight" means -> notable-top-25%. As stated in our emails, we are disassociating presentation type from AC recommendations

**Summary Of Ac-Reviewer Meeting:**

The author-reviewer discussion and further evaluation of the paper were enough to make a definite decision.